# Evaluation of Nanotargeted ^111^In-Cyclic RGDfK-Liposome in a Human Melanoma Xenotransplantation Model

**DOI:** 10.3390/ijms22031099

**Published:** 2021-01-22

**Authors:** Si-Yen Liu, Sheng-Nan Lo, Wan-Chi Lee, Wei-Chuan Hsu, Te-Wei Lee, Chih-Hsien Chang

**Affiliations:** 1Division of Isotope Application, Institute of Nuclear Energy Research, Taoyuan 32546, Taiwan; b9029506@yahoo.com.tw (S.-Y.L.); loshengnan@iner.gov.tw (S.-N.L.); leewc@iner.gov.tw (W.-C.L.); wchsu@iner.gov.tw (W.-C.H.); tewei123456@gmail.com (T.-W.L.); 2Department of Biomedical Imaging and Radiological Sciences, National Yang-Ming University, Taipei 11221, Taiwan

**Keywords:** ^111^In-cyclic RGDfK-liposome, α_V_β3 integrin, Nanotargeted liposome

## Abstract

Nanotargeted liposomes may be modified with targeting peptide on the surface of a prepared liposome to endow specificity and elevate targeting efficiency. The aim of this study was to develop a radioactive targeted nanoparticle, the ^111^In-cyclic RGDfK-liposome, and its advantage of recognizing the α_V_β3 integrin was examined. The cyclic RGDfK modified liposomes were demonstrated the ability to bind the α_V_β3 integrin expressed on the surface of human melanoma cell in vitro and in vivo. The effects of the cyclic RGDfK-liposome on the functioning of phagocytes was also examined, showing no considerable negative effects on the engulfment of bacteria and the generation of reactive oxygen species. Based upon these findings, the cyclic RGDfK- liposome is said to be a promising agent for tumor imaging.

## 1. Introduction

Liposome is one of nanomedicines originally designed to improve the distribution and target site accumulation of systemically administered therapeutic agents [1,2,3,4]. They are spherical, self-closed formed lipid bilayer with phospholipids in which entrapped radionuclides [5] or pharmaceuticals [6]. These nanosized drug delivery systems with some modifications, such as PEGylated liposome [7,8], are popularly used for increased absorption, delayed excretion, decreased uptake and removal from circulation by reticuloendothelial system, longer half-life within blood circulation and lower toxicity. Besides, liposomes could accumulate in tumor through tumor vasculature leaky and enhanced permeability and retention (EPR) effect [6,9]. Indium-111 (^111^In) (t_1/2_ = 2.81 days, 172 and 247 keV characteristic X-ray emission), is a commonly used radionuclide for single-photon emission computed tomography (SPECT) imaging. Harrington and his colleague also report about the biodistribution and SPECT/CT imaging of ^111^In-DTPA-labeled PEGylated liposomes in advanced cancer patients [10]. However, the quality of generated images was not high enough to be used in patient selection and treatment evaluation.

Tumor cells use cell surface adhesion molecules to mediate the complex cell binding activity required for tumor growth and metastasis [11]. Among adhesion molecules, the α_V_β3 integrin is of particular importance in tumor progression because that α_V_β3 integrin is highly expressed on activated endothelium in malignant tissues and that ligation of α_V_β3 integrin plays a role in survival and maturation of newly forming blood vessels for tumor angiogenesis [12]. Hence, the α_V_β3 integrin has been recognized a principle angiogenic marker in the progression of tumor [12,13]. With its essential function for tumor progression, the α_V_β3 integrin had been aimed at cancer detection by molecular imaging modalities [14,15,16,17]. This integrin can recognize extracellular matrix proteins, containing arginin-glycine-aspatic acid (RGD) sequence therein; however, only cyclic RGD peptides of being conformational restriction do selectively achieve to discriminate different integrin subtypes, and this group of RGD peptides is said to be an ideal molecule of being used to deliver drugs to cancer cells [8,18,19].

Although PEGylated liposome can deliver drug to tumor sites through enhanced permeability and retention effects [18,19,20], it still lacks enough molecular specificity for tumor targeting. Hence, surface-modified functional liposome is becoming increasingly popular for diagnostic and therapeutic applications [3]. Liposomes may be modified with targeting peptide RGD by either coupling RGD on the surface of a prepared liposome or mixing phospholipids with RGD phospholipids and make them self-assemble into liposomes to endow specificity and elevate targeting efficiency [21]. In this study, a radioactive targeted ^111^In-cyclic RGDfK-liposome was prepared, molecular targeted imaging and efficacy of the nanoparticle for tumor imaging was evaluated.

## 2. Results

### 2.1. The Cyclic RGDfK-Liposome Inhibits the α_V_β_3_ Integrin-Mediated Cell Adhesion of Human Melanoma Cells To Extracellular Matrix Proteins

To generate the cyclic RGDfK-liposome, the peptide was first conjugated to NHS-PEG-DSPE to generate PEG-linked-peptide conjugates; the conjugates were inserted into conventional liposome (Figure 1A). Next, we tested the effect of the cyclic RGDfK-liposome on the α_V_β3 integrin -mediated cellular adhesion using human malignant melanoma A375.S2 cells on recombinant extracellular matrix proteins. We found that treatment with 100 nM cyclic RGDfV or cyclic RGDfK peptide markedly reduced A375.S2 cell adhesion to fibronectin, collagen I, and collagen IV, indicating that A375.S2 cells adherent to recombinant extracellular proteins was accurately mediated by the α_V_β3 integrin (Figure 1B). Under the same experimental conditions, when compared to control and liposome-treated groups, cyclic RGDfK-conjugated liposome showed a considerable inhibitory effect on A375.S2 cell bond to either fibronectin or collagen I (Figure 1B). The expression levels of the α_V_β3 integrin were also examined by FITC mouse anti-human CD51/CD61 monoclonal antibody, and the data showed that the α_V_β3 integrin intensity was similar among experimental groups (Figure 1C). Taken together, these findings indicated that the conjugating process did not abolish the cell-targeting efficiency of the cyclic RGDfK-conjugated liposome. The inhibition effect on the α_V_β3 integrin-mediated cellular adhesion was not due to different expression levels of the integrin but because of the specificity of the cyclic RGDfK-conjugated liposome at targeting human melanoma cells.

### 2.2. In Vivo Characterization of Imaging Efficiency of the Cyclic RGDfK-Liposome

In addition to its in vitro targeting profile, the cyclic RGDfK-liposome was then tested for its in vivo targeting profile in a nude mouse human melanoma cells xenotransplantation model, so the radioactive cyclic RGDfK-liposome was prepared. The ^111^In-labeled-liposome or the ^111^In-labeled cyclic RGDfK-liposome was injected intravenously into the mice with human melanoma cells, and mice were subjected to single photon emission computed tomography/micro-computed tomography analysis, and images were captured at 24 h post-injection; the experimental scheme was showed in Figure 2A. In contrast to the mice receiving the ^111^In-labeled liposome, as showed in Figure 2B, the mice receiving the ^111^In-labeled cyclic RGDfK-liposome showed a clear nuclear image of tumor nodules due to the accumulation of the ^111^In-labeled cyclic RGDfK-liposome; furthermore, the specificity of the ^111^In-labeled cyclic RGDfK-liposome of being targeted at tumors was confirmed by mice receiving the cyclic RGDfV peptide (1 mg/kg) before having an injection of the ^111^In-labeled cyclic RGDfK-liposome, the data showed, when compared to the mice without receiving the cyclic RGDfV peptide, that the radioactive signal within tumors in mice receiving the same peptide did decrease (Figure 2B, b and c panels). Additionally, with regions of interest analysis of tumor nodules, it showed that the tumor-to-background ratio in the mice with an injection of the ^111^In-labeled cyclic RGDfK-liposome was significantly higher than the ratio in the mice with the ^111^In-labeled liposome and that the tumor-to-background ratio in mice with an injection of the ^111^In-labeled cyclic RGDfK-liposome but with the cyclic RGDfV peptide treatment was, however, significantly lower than the ratio in the mice with the same injection but without the cyclic RGDfV peptide (Figure 2C).

The tissue uptake of the two types of the radioactive liposome was analyzed, not least tumors and blood; the uptake of the ^111^In-labeled cyclic RGDfK-liposome in tumors and in blood was 5.3% ID/g and 1.1%ID/g, of the ^111^In-labeled liposome in both tissue, 2.2% ID/g for tumors, 2.1% ID/g for blood, and when compared to the tumor-to-blood ratio in the mice receiving the ^111^In-labeled-liposome (10.04), the ratio in the mice receiving the ^111^In-labeled-cyclic RGDfK-liposome significantly increased (4.8) (Table 1).

Another mouse model, on the other hand, of having a metastatic growth of human melanoma cells, was also used to evaluate the efficacy of the ^111^In-labeled cyclic RGDfK-liposome of being targeted at tumors, and the experimental scheme was showed in Figure 3A; we found, as showed in Figure 3B, in contrast to the mice with an injection of the ^111^In-labeled liposome, of the micro-metastasis site found in jejunal lymph node, the mice with an injection of the ^111^In-labeled cyclic RGDfK-liposome showed of having a clear nuclear image of tumor nodules due to the accumulation of the ^111^In-labeled cyclic RGDfK-liposome, which was further confirmed by dissecting the tumors found in jejunal lymph nodes and analyzing the radioactivity of ^111^In therein (Figure 3B, of the bottom of each panels and Table 2), finding that the uptake in tumors and in blood was 6.2%ID/g and 1.1%ID/g. On the other hand, the radioactivity in tumors and in blood from the mice receiving the ^111^In-labeled liposome was 2.9% ID/g and 2.1% ID/g, respectively. The cellular composition of tumor nodules dissected from jejunal lymph nodes was confirmed by a FITC mouse anti-human CD51/CD61 monoclonal antibody stain on the cells, and as showed in Figure 3C, the cells isolated from tumors expressed the α_V_β3 integrin, and when compared to the tumor-to-blood ratio in the mice receiving the ^111^In-labeled-liposome (1.3), the ratio in the mice receiving the ^111^In-labeled-cyclic RGDfK-liposome significantly increased (5.6) (Table 2). Taken together, these findings clearly indicated that with the radioactive cyclic RGDfK-liposome, human melanoma cells could be depicted.

### 2.3. The Cyclic RGDfK-Liposome Has No Considerable Immunogenicity

We also tested whether the cyclic RGDfK-liposome alters the phagocytotic capacity of mouse macrophage, RAW 264.7 cell. The flow cytometry-based system was performed to determine a ratio between cells and the opsonized fluorescence-labeled *Escherichia coli* (*E. coli*), the ratio 1:10 was the optimal condition for phagocytosis assay (Figure 4A). RAW 264.7 cells were treated with liposome or the cyclic RGDfK-liposome, followed by incubating with *Escherichia coli* (*E. coli*), Alexa Fluor^®^ 594 conjugate. In the presence of the cyclic RGDfK-liposome (10 or 100 nM) (Figure 4B, d to f panels), though that high concentration of the cyclic RGDfK-liposome slightly reduced the cell’s ability to phagocytosis, this inhibition was comparable to the liposome-treated group (Figure 4B, a–c panels and d to f panels). The production of reactive oxygen species is accompanied by phagocytosis; therefore, the generation of reactive oxygen species was examined whether the cyclic RGDfK-liposome has any effect on it within phagocyte, not least in reaction to lipopolysaccharide, a well-known potent stimulator for reactive oxygen species production. In the absence of lipopolysaccharide, the effect of the cyclic RGDfK-liposome on the production of reactive oxygen species was comparable to the liposome-treated group (Figure 4C, a and b panels). By contrast, in the presence of lipopolysaccharide, it showed that liposome inhibited the phagocyte’s ability to produce reactive oxygen species (Figure 4D, panel b); however, the cyclic RGDfK-liposome in different concentrations did not inhibit the ability in response lipopolysaccharide stimulation to make reactive oxygen species (Figure 4D, a and b panels).

In addition to cell-based assays, whether there was difference between the accumulation of the ^111^In-labeled liposome and of the ^111^In-labeled cyclic RGDfK-liposome in mice’s lymph nodes was also examined. The mice without human melanoma were injected with radioactive liposome and sacrificed 24 h after the administration to collect lymph nodes, including jejunal, inguinal and axillary lymph node, and blood samples; then, the radioactivity of ^111^In therein to be measured. The data showed both the radioactivity in lymph nodes and in blood, as well as each one of the radioactivity proportion of lymph node found in the mice receiving the ^111^In-labeled-cyclic RGDfK-liposome were comparable with those found in the mice receiving the ^111^In-labeled-liposome (Table 3). These findings indicated the cyclic RGDfK- liposome was a targeted liposome of inconsiderable immunogenicity.

## 3. Discussion

Liposome in nanotechnology provides a highly multipurpose platform for exploring several different approaches, thereby improving the drug delivery for targeting tumor cell or the imaging of tumors [2,22,23,24,25,26]. Although PEGylation can enhance the stability of liposome and increase the blood circulation time of liposome, it reduces the strength of liposome to interact with target cells, resulting in inappropriate cellular uptake and poor endosomal escape [27,28,29,30]. With the result that, surface-modified liposome is becoming increasingly popular, and several ligands were added at the ends of PEG moieties to generate targeted PEGylation liposome [19,28,31,32,33].

The arginin-glycine-aspatic acid (RGD) sequence is the cell attachment site of a large number of adhesive extracellular matrix, blood, and cell surface proteins, and nearly half of the over 20 known integrins recognize this sequence in their adhesion protein ligands. The integrin-binding selective activity to only one or a few of the RGD-directed integrins can be designed by cyclizing peptides with selected sequences around the RGD. The cyclic peptide cRGDfK and cRGDfV showed high binding affinity for integrin α_V_β_3_, but only cRGDfK peptide acts as vectors for delivery of chemotherapeutics [34,35]. In this present work, cRGDfV peptide was employed as positive control in cell adhesion assay and as potent competitor in human melanoma cells xeno-transplantation animal model to confirm the preparing process of the cyclic RGDfK-liposome did not abolish the α_V_β_3_ integrin-binding selective activity of the cRGDfK peptide.

Cells in malignant tumors can invade nearby tissues and spread out the original site in the body through blood vessels or lymphatic vessels via adhesion molecules. Expression of integrin on melanoma cells is said not only to assist extracellular matrix anchorage-dependent tumor cell proliferation but to support extracellular matrix anchorage-independent tumor cell invasion [34,35]. In this present work, the α_V_β3 integrin was employed as target receptor for its highly expression level on tumor tissues [12,15], and two types of animal models, of which one was human melanoma cells xeno-transplantation model and the other was human melanoma cells xeno-transplantation model with spontaneous micro-metastasis confirmed by isolating the cells from tumor nodules that were found mainly in the mice’s jejunal lymph node to examine the expression levels of the human α_V_β3 integrin, were used to evaluate the capability of the ^111^In-labeled cyclic RGDfK-liposome to distinguish human melanoma cells.

Since several studies had demonstrated that being capable of recognizing disease-related macromolecules is one of the advantages of targeted nano-particles, in this study, when compared to the ^111^In-labeled liposome, clear images of human melanoma at which the ^111^In-labeled cyclic RGDfK-liposome targeted in mice captured by Nano SPECT/CT, which was parallel to the finding of regions of interest analysis of images for radioactivity in tumor nodules, to the results of radioactivity measurement of tumor samples, and to the proportion of the radioactivity in tumor samples to the radioactivity in blood samples: all indicated that the cyclic RGDfK-liposome had the advantage of being more specific and precise to be targeted at tumor cells expressing the α_V_β3 integrin than the non-targeted liposome used in this study. Despite the efficacy, in general, studies for potential toxicity of the cyclic RGDfK-liposome are required. Most important, the cyclic RGDfK-liposome should keep not interacting with the immune system [36].

Phagocytes are the main cellular components of innate immunity in the immune system, which professionalize engulfing microbes, as well as foreign molecules, and destroy them by enzymatic digestion and reactive oxygen species [35], yet these reactions can be affected at different levels [27,37] because, in this study, the surface of the conventional PEGylaiton liposome was modified to add ligands; however, whether this modification can elicit any undesirable response, like immunostimulation or immunosuppression, largely due to the composition of the PEGylation liposome’s surface being altered, remained unclear [38,39,40]. Hence, we set a flow cytometry-based system to examine whether the cyclic RGDfK-liposome had any effect on the function of phagocytes. We found that this formulation had no considerable inhibition effect on engulfing bacteria in phagocytes, nor had considerable inhibition effect on the production of reactive oxygen species in phagocytes. Additionally, the distribution of the cyclic RGDfK-liposome to lymph nodes being comparable with the non-targeted liposome in this study indicated that the cyclic RGDfK-liposome was a targeted liposome without considerable immunogenicity [38].

## 4. Materials and Methods

### 4.1. Cells

A375.S2 human malignant melanoma cell line (stock number: BCRC 60263) and RAW 264.7 murine macrophage cell line (stock number: BCRC 60001) were obtained from Bioresource Collection and Research Center, Taiwan. A375.S2 cells were maintained in Minimum essential medium with 2 mM L-glutamine, 0.1 mM non-essential amino acids, 1 mM sodium pyruvate, 1.5 g/L sodium bicarbonate and 10% heat-inactivated fetal bovine serum. RAW 264.7 cells were cultured in Dulbecco’s modified Eagle’s medium with 4 mM L-glutamine, 4.5 g/L glucose, 1.5 g/L sodium bicarbonate and 10% heat-inactivated fetal bovine serum. The cells were cultured under an atmosphere of 5% CO_2_ at 37 °C.

### 4.2. Reagents and Antibodies

Polyethylene glycol PEG_2000_-carbamyl distearoylphosphatidyl ethanolamine (NHS-PEG-DSPE) was purchased from the NOF Corporation, Japan. Cyclic RGDfK peptide was obtained from Peptides International, Inc. (Louisville, KY, USA). The pegylated liposome (Nano-X) (∼100 nm diameter) composed of 1,2-distearoyl-sn-glycero-3-phosphocholine (DSPC), Cholesterol, and DSPE-PEG_2000_ (molar ratio is 3:2:0.3) was provided by Taiwan Liposome Company (Taipei, Taiwan). FluoroProfie^®^ Protein Quantification Kit was from Sigma (St. Louis, MO, USA), CytoSelect TM Cell Adhesion Assay Kit from CELL BIOLABS, Inc. (San Diego, CA, USA). Escherichia coli BioParticles^®^, Alexa Fluor^®^ 594 conjugate and Escherichia coli BioParticles^®^ opsonizing reagent were obtained from Invitrogen (Carlsbad, CA, USA). Fluorescein isothiocyanate (FITC)-labeled mouse anti-human CD51/CD61 monoclonal antibody and FITC-labeled mouse IgG1 isotype control monoclonal antibody were purchased from BD Biosciences (San Jose, CA, USA).

### 4.3. Preparing the Cyclic RGDfK-Liposome

The cyclic RGDfK peptide reacted with NHS-PEG-DSPE at molar ratio 1:1.1 in N,N-dimethylformamid and incubated at room temperature for 24 h. The theory molecular weight of cyclic RGDfK-PEG-DSPE conjugate was examined by matrix assisted laser desorption/ionization-time of flight mass spectrometry (MALDI-TOF/MS), using acetonitril:water = 1:1 with 0.1% trifluoroacetate as the matrix solution, supplied with 10mg/mL α-cyano-4-hyroxy-cinnamic acid. To prepare cyclic RGDfK-conjugated liposome, 0.5 mg cyclic RGDfK-PEG-DSPE conjugate was incubated with 0.5 mL liposome solution at 60 °C for 30 min. The peptide insertion efficacy was determined by FluoroProfie^®^ Protein Quantification Kit and performed according to manufacturer’s protocol. The preparation for the ^111^In-labeled liposome was performed on the basis of ^111^In-oxine entrapment method [41].

### 4.4. Animals and Single Photon Emission Computed Tomography/Micro-Computed Tomography Imaging

Female, BALB/c AnN. Cg-Foxn1^nu^/Cr1Nar1 mice were obtained from National Laboratory Animal Center, Taiwan. Nude mice were injected subcutaneously with 2 × 10^5^ A375.S2 human melanoma cells in their necks. Two weeks after the injection, animals had developed nodules of similar size, about 2 mm in diameter. For taking nuclear images with NanoSPECT/CT^®plus^ scanner system (NanoSPECT/CT PLUS, Mediso, Alsotorokvesz, Budapest, Hungary), the mice was injected with the ^111^In-labeled liposomes (50 μCi), and the images of mice was captured 24 h after an injection of radioactive liposomes, during which the mice were anesthetized with 1.5% isoflurine by a ventilator.

### 4.5. Cell Adhesion Assay

Adhesion assay were performed with CytoSelect TM Cell Adhesion Assay Kit. Principle, A375.S2 cells (1 × 10^6^/well) in working solution (serum-free MEM medium, 1mM Mn^2+^, and 0.5% bovine serum albumin) were allowed to adhere to extracellular matrix proteins-coated plate for 90 min at 37 °C. Cells were analyzed for binding in the presence or absence of test subjects. Non-adherent cells were removed by a flick wash. The adherent cells were stained with crystal violet and examined the optical density at λmax 560 nm.

### 4.6. Measuring the αVβ3 Integrin Expression

The cells (1 × 10^6^/mL) was incubated with 1 μg FITC-labeled mouse anti-human CD51/CD61 or FITC-labeled mouse IgG1κ isotype control monoclonal antibody at 4 °C for 60 min, followed analyzing by flow cytometry.

### 4.7. Phagocytosis Assay

The opsonization of fluorescent-labeled Escherichia coli BioParticles was performed according to manufacturer’s protocol. The RAW 264.7 cells were co-incubated with opsonized fluorescently labeled Escherichia coli BioParticles (ratio = 1:10) in Dulbecco’s modified Eagle’s medium containing 10% heat-inactivated fetal bovine serum at 37 °C for 1 h. The fluorescence of extracellular bound particles was quenched by adding tryphan blue, followed analyzing by flow cytometry.

### 4.8. Reactive Oxygen Species Production Assay

RAW 264.7 cells (1 × 10^6^ cells/mL) were co-incubated with 100 g dichlorofluorescin diacetate in 0.1 mL sterile phosphate buffered saline at 37 °C for 1 h in the presence or absence of liposome or cyclic RGDfK-functionalized liposome. For lipopolysaccharide stimulation group, RAW 264.7 cells (1 × 10^6^ cells/mL) were co-incubated with 200 μg lipopolysaccharide in 0.1 mL sterile phosphate buffered saline at 37 °C for 1 h in the presence or absence of liposome or cyclic RGDfK-functionalized liposome with 100 μg dichlorofluorescin diacetate. The testing samples were analyzed by flow cytometry.

### 4.9. Statistical Analyses

The data were obtained from at least three independent experiments and analyzed with Student’s *t*-test. Data are means ± SDs. Significance was set at *p* < 0.05.

## 5. Conclusions

We report herein on the successful design of modifying the surface of conventional liposome to add the cyclic RGDfK peptide for the targeted delivery of the radioactive cyclic RGDfK liposome to human melanoma cells, where the α_V_β_3_ integrin was expressed.

## Figures and Tables

**Figure 1 ijms-22-01099-f001:**
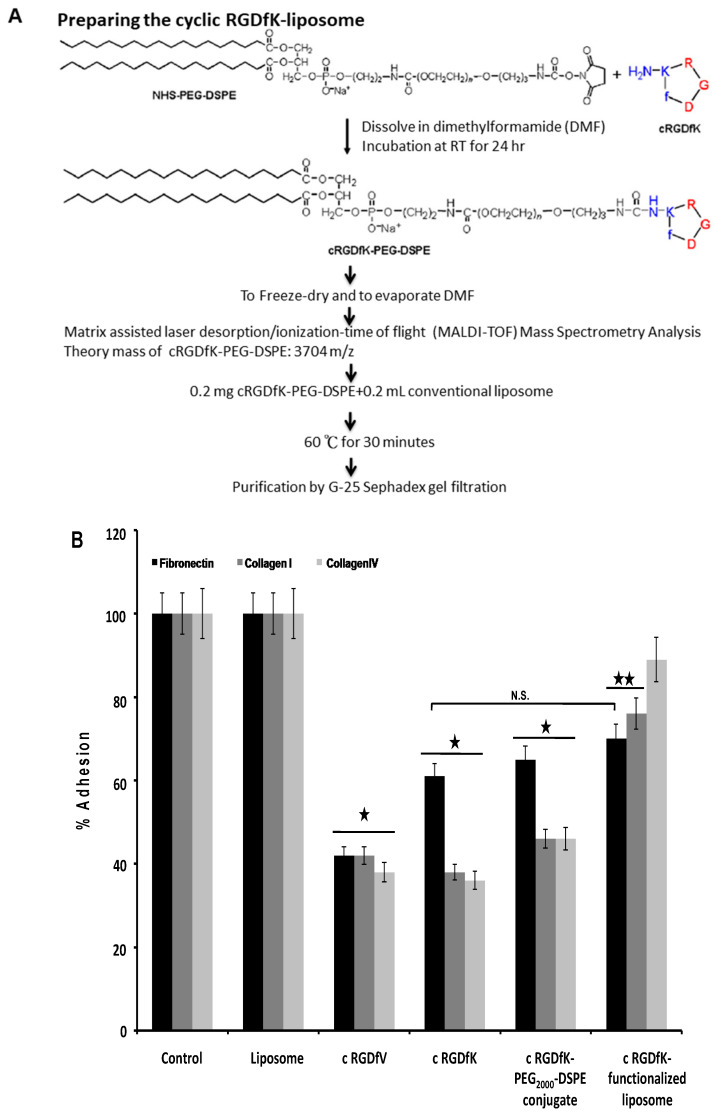
Evaluating the effect of the cyclic RGDfK-liposome on α_V_β_3_ integrin-mediated cell adhesion. (**A**) The cyclic RGDfK-liposome preparation was showed. A375.S2 cells was pretreated with 100 nM cRGDfV, cRGDfK, cRGDfK-PEG_2000_-DSPE or the cRGDfK-liposome and allowed to adhere to a coating of fibronectin, collagenI or collagen IV for 1 h at 37 °C before washing and detection of cell adhesion (**B**) or followed by staining with FITC-labeled mouse anti-human CD51/CD61 monoclonal antibody (**C**). The black star indicated *p* < 0.01 and duplicate black stars for *p* < 0.05, when compared with liposome group; N.S., non-significant. These experiments were performed for five times.

**Figure 2 ijms-22-01099-f002:**
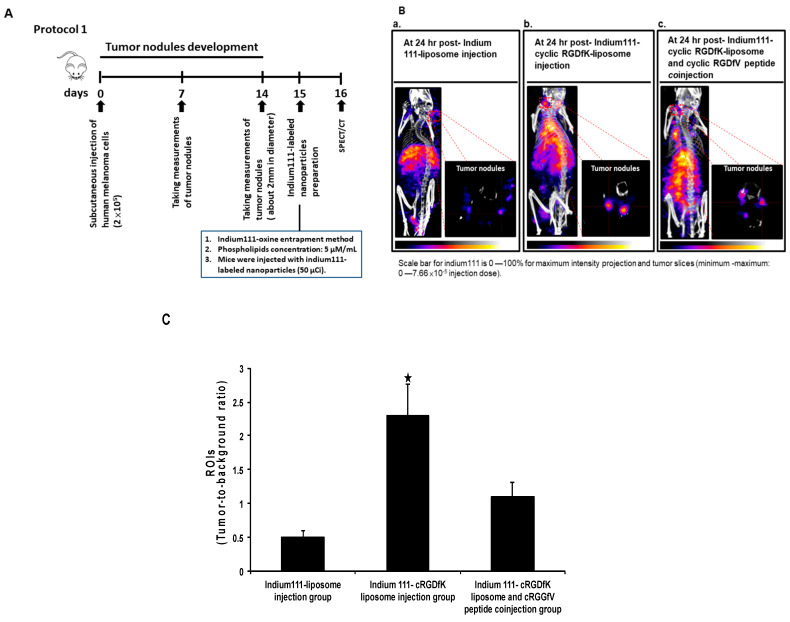
Non-invasive imaging with Nano SPECT/CT in mice without a metastatic growth of human melanoma cells. (**A**) Experimental scheme; (**B**) The nude mice with human melanoma were injected with the ^111^In-liposome (panel a), with the ^111^In-labeled-cyclic RGDfK-liposome (panel b), or with an injection of being a combination of the ^111^In-labeled-cyclic RGDfK-liposome and the cyclic RGDfV peptide (panel c), and images was captured at 24 h post-radioactive liposomes injection. (**C**) The region of interest analysis of tumors was showed. The black star indicated *p* < 0.01, when compared to the other groups. The figure showed here was a representative example of three independent experiments in which each group consisted of three mice.

**Figure 3 ijms-22-01099-f003:**
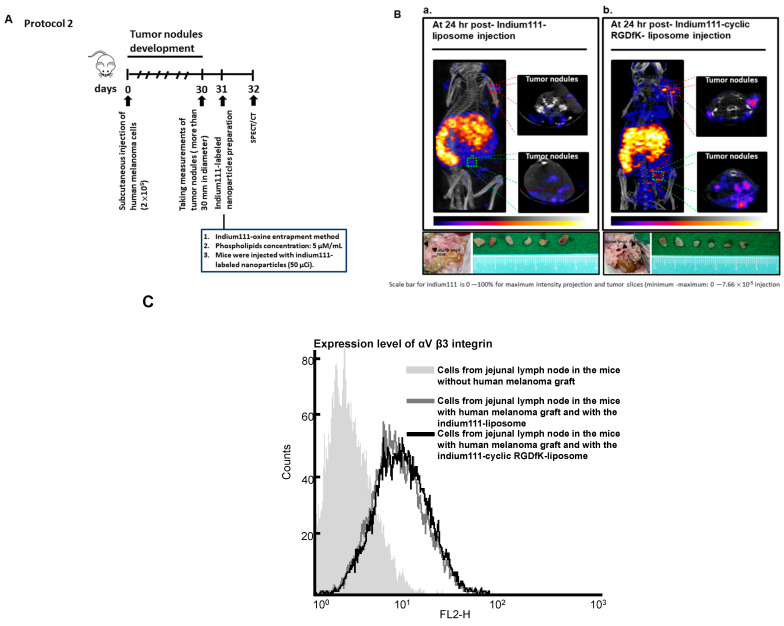
Non-invasive imaging with Nano SPECT/CT in mice with a metastatic growth of human melanoma cells. (**A**) Experimental scheme; (**B**) The nude mice with human melanoma were injected with the ^111^In-liposome (panel a) and with the ^111^In-labeled-cyclic RGDfK-liposome (panel b), and images was captured at 24 h post- radioactive liposomes injection. The tumors found around jejunal lymph node were dissected. (**C**) The measurement of αVβ3 integrin expression by flow cytometry. The figure showed here was a representative example of three independent experiments in which each group consisted of three mice.

**Figure 4 ijms-22-01099-f004:**
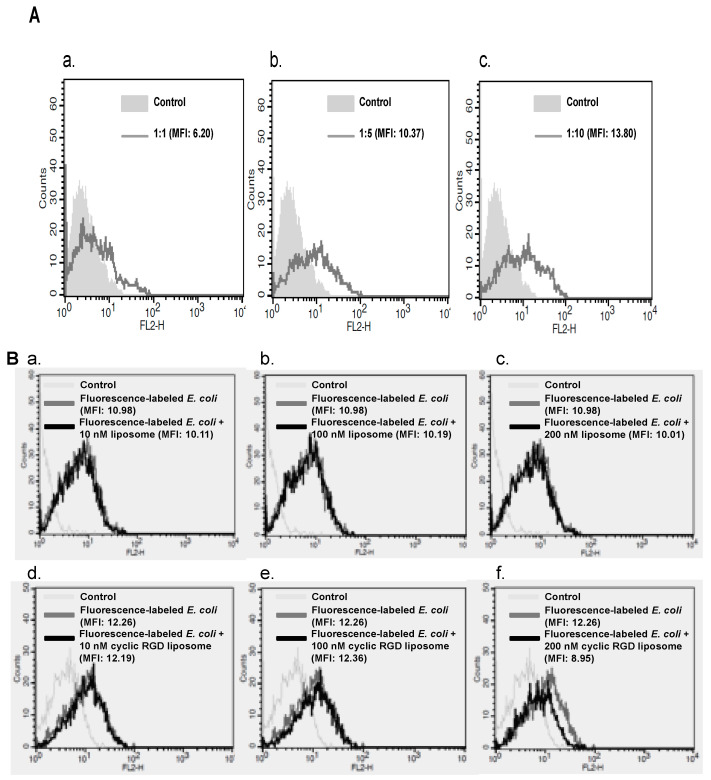
Evaluating the effect of the cyclic RGDfK-liposome on functioning of phagocytes. (**A**) Determining the optimal ratio of RAW 264.7 cells to the opsonized fluorescent-labeled *Escherichia coli* (*E. coli*). (**B**) The measurement of the fluorescent-labeled *Escherichia coli* (*E. coli*) engulfed in RAW264.7 cells in the presence of liposome (panels a–c) or the cyclic RGDfK-liposome (panels d–f). (**C**) The measurement of reactive oxygen species generation in RAW264.7 cells in the presence of liposome (panel a) or the cyclic RGDfK-liposome (panel b) (**D**) Measuring the effect of liposome or the cyclic RGDfK-liposome on reactive oxygen species generation in RAW264.7 cells with low concentration (panel a) or high concentration (panel b) in the presence of lipopolysaccharide (LPS). These experiments were performed five times.

**Table 1 ijms-22-01099-t001:** The percentage of injection dose of the ln-111 cyclic RGDfK-liposome or the ln-111 liposome in the tissue samples and the radioactivity proportion of tumor to blood in the mice without a metastatic growth of human melanoma cells.

	^111^ln-Cyclic RGDfK-Liposome	^111^ln-Liposome
**t** **umor (%ID/g)**	5.3 ± 00.09	2.2 ± 00.07
**bl** **ood (%ID/g)**	1.1 ± 0.1	2.1 ± 00.02
**t** **umor-to-** **b** **lood ratio**	4.8	1

**Table 2 ijms-22-01099-t002:** The percentage of injection dose of the ^111^ln-cyclic RGDfK-liposome or the ^111^ln-liposome in the tissuesamples and the radioactivity proportion of tumor to blood in the mice with a metastatic growth of human melanomacells.

	^111^ln-Cyclic RGDfK-Liposome	^111^ln-Liposome
**t** **umor (%ID/g)**	6.2 ± 00.09	2.9 ± 00.09
**b** **lood (%ID/g)**	1.1 ± 00.05	2.1 ± 0.1
**t** **umor-to-** **b** **lood ratio**	5.6	1.3

**Table 3 ijms-22-01099-t003:** The percentage of injection dose per gram (%ID/g) of the ^111^ln-cyclic RGDfK-liposome or the ^111^ln-liposome in lymph nodes and the radioactivity proportion of lymph nodes to blood in the mice without human melanoma xenotransplantation.

	Drug	^111^ln-Cyclic RGDfK-Liposome	^111^ln-Liposome
Item			
**jejunal lymph node (%ID/g)**		2.1 ± 0.1	2.3 ± 00.09
**inguinal lymph node (%ID/g)**		0.5 ± 00.01	0.7 ± 0.1
**axillary lymph node (%ID/g)**		0.7 ± 0.12	1.2 ± 00.09
**blood (%ID/g)**		1.1 ± 00.05	1.3 ± 00.07
**lymph node-to-blood ratio**	**jejunal to blood**	1.9	1.7
	**inguinal to blood**	0.45	0.5
	**axillary to blood**	0.6	0.9

## Data Availability

Not applicable.

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
