# Peer review of "Evaluation of Nanotargeted 111In-Cyclic RGDfK-Liposome in a Human Melanoma Xenotransplantation Model"

_ijms, 2021, doi:10.3390/ijms22031099_

Round 1

Reviewer 1 Report

The following comments may help to improve the manuscript.

(1) In Table 1, the tumor-to-blood ratio of 111In-liposomes should be approximately 1, not 10.

(2) Could authors please provide the rationale of including both cRGDfK and cRGDfV in the studies? A brief discussion may reduce the possibility of confusing some readers who are not familiar with cRGD peptides.  

(3) Do 111In-cRGDfK-liposomes have a high accumulation in other major organs such as liver and spleen (Figure 2B and 3B)? 

(4) cRGD peptides can target fast-growing vessels and how does the time after tumor-inoculation affect the targeting effect of cRDG-conjugated liposomes? Have authors compared the targeting/imaging of cRGD-liposomes administered on 15th and 31th day?  

Author Response

Dear Reviewer:

We would like to re-submit a revised article entitled “Evaluation of Nanotargeted 111In-cyclic RGDfK-liposome in a human melanoma xenotransplantation model” toInternational Journal of Molecular Sciences”. We have revised the manuscript according to your comments and suggestions.

We think this work is worthwhile for publishing in International Journal of Molecular Sciences. We are looking forward to hearing you soon.

Reviewer 1

Comments and Suggestions for Authors

The following comments may help to improve the manuscript.

  • In Table 1, the tumor-to-blood ratio of 111In-liposomes should be approximately 1, not 10.

Ans: Thank you very much. The tumor-to-blood ration of 111In-liposomes showed in Table 1 had been corrected.

  • Could authors please provide the rationale of including both cRGDfK and cRGDfV in the studies? A brief discussion may reduce the possibility of confusing some readers who are not familiar with cRGD peptides.

Ans: Thank you for suggestion. We have added the rationale for the study of both cRGDfK and cRGDfV in second paragraph of the discussion section.

  • Do 111In-cRGDfK-liposomes have a high accumulation in other major organs such as liver and spleen (Figure 2B and 3B)? 

Ans: Many thanks. In the experimental groups showed in Figure 2B and 3B, when compared to the 111In-liposomes injecting group, we did not found 111In-cRGDfk-liposomes have a higher accumulation in liver and spleen.

  • cRGD peptides can target fast-growing vessels and how does the time after tumor-inoculation affect the targeting effect of cRDG-conjugated liposomes? Have authors compared the targeting/imaging of cRGD-liposomes administered on 15th and 31th day? 

Ans: Thank you so much. We had compared the targeting/imaging of cRGD-liposomes administered on 15th and 31th day, and it showed that cRGD-liposomes were majorly accumulated in liver.

Reviewer 2 Report

The manuscript is very well organized and the images are very interesting. It is recommended to increase the size of Fig 1A to clear understand the scheme and to increase Fig 2B and 3B, in 3C please distinguish better the lines, as well as in Fig 4. Revise the bacteria names and some "in vivo" lacking the italic format.

Please clarify how the unconjugated peptide is removed from the liposome suspension, and please include information on the size and surface charge of the liposomes.

An English revision is also recommended prior acceptance for publication.

Author Response

Dear Reviewer:

We would like to re-submit a revised article entitled “Evaluation of Nanotargeted 111In-cyclic RGDfK-liposome in a human melanoma xenotransplantation model” toInternational Journal of Molecular Sciences”.  We have revised the manuscript according to reviewer’s comments and suggestions.

We think this work is worthwhile for publishing in International Journal of Molecular Sciences. We are looking forward to hearing you soon.

Reviewer’s comments and suggestions for authors

The manuscript is very well organized and the images are very interesting. It is recommended to increase the size of Fig 1A to clear understand the scheme and to increase Fig 2B and 3B, in 3C please distinguish better the lines, as well as in Fig 4. Revise the bacteria names and some "in vivo" lacking the italic format.

Please clarify how the unconjugated peptide is removed from the liposome suspension, and please include information on the size and surface charge of the liposomes.

An English revision is also recommended prior acceptance for publication.

Ans:

(1) The size of Fig. 1A, 2B, 3B, 3C and Fig.4 all had been increased.

(2) Bacteria names had been revised, and the format of “in vivo” had been revised to “in vivo”.

(3) The unconjugated peptide was removed by G25-Sephadex gel filtration.

(4) We have added the size of NanoX liposome in materials and methods section. According the previous report, the NanoX liposome has natural surface charge.
